# Coronary Artery Disease in Patients Older than 35 and Eligible for Cardiovascular Secondary Prevention: An Italian Retrospective Observational Analysis of Healthcare Administrative Databases

**DOI:** 10.3390/jcm10204708

**Published:** 2021-10-14

**Authors:** Silvia Calabria, Giulia Ronconi, Letizia Dondi, Carlo Piccinni, Enrico Cinconze, Antonella Pedrini, Immacolata Esposito, Alice Addesi, Nello Martini, Aldo Pietro Maggioni

**Affiliations:** 1Fondazione Ricerca e Salute (ReS)—Research and Health Foundation, Casalecchio di Reno, 40033 Bologna, Italy; ronconi@fondazioneres.it (G.R.); dondi@fondazioneres.it (L.D.); piccinni@fondazioneres.it (C.P.); enrico.cinco@libero.it (E.C.); pedrini@fondazioneres.it (A.P.); nello.martini@libero.it (N.M.); maggioni@anmco.it (A.P.M.); 2Drugs & Health srl, 00187 Rome, Italy; esposito@drugs-health.it (I.E.); addesi@drugs-health.it (A.A.); 3ANMCO Research Center, Fondazione per il Tuo cuore, 50121 Florence, Italy

**Keywords:** coronary artery disease, secondary prevention, public health, database

## Abstract

Background: This study describes patients with coronary artery disease (CAD) who are eligible for secondary prevention and assesses their healthcare consumption and costs from the perspective of the Italian National Health Service (INHS). Methods: From the Fondazione Ricerca e Salute’s database, which collects Italian healthcare administrative data, all patients aged ≥ 35, with ≥1 primary in-hospital CAD diagnosis and/or procedure on the coronary arteries, or with the specific disease exemption code, and who are suitable for long-term secondary prevention treatments, were identified in 2018 and analyzed. Demographics, comorbidities, one-year supplied drugs, hospitalizations, and costs were analyzed. Results: From >3 million inhabitants aged ≥ 35, 46,063 (1.3%) were identified (72.1% males, mean age 70 ± 12; approximately 50% with ≥3 comorbidities). During a one-year follow-up, 96.4% were treated with ≥1 drug for secondary prevention (mainly antiplatelets and lipid lowering agents), 69.4% with ≥1 concomitant cardiovascular drug, and 95.8% with ≥1 concomitant non-cardiovascular therapy. Within one year, 30.6% of patients were hospitalized at least once, mostly due to non-cardiovascular events. Calculated by mean, the INHS paid EUR 6078 per patient. Conclusions: This analysis confirms the relevant burden of CAD for patients with many comorbidities and who are frequently hospitalized, and the burden on the INHS. A multidisciplinary healthcare approach is encouraged to improve patients’ outcomes and reduce costs for the INHS.

## 1. Introduction

Although the prevalence and mortality rates of coronary artery disease (CAD) had decreased by the early 1980s in Europe, they continue to remain high, with 9.2% of the European population affected by CAD and more than 1,700,000 patients dying every year [1,2]. CAD can be the consequence of long-term atherosclerotic disease, but its dynamic progression can unexpectedly lead to adverse events, such as myocardial infarction (MI), and stroke and cardiovascular (CV) death [1,3]. CAD mortality and hospitalization rates could be halved by modest risk factor reductions [2]. The slowdown of CAD progression is often more effective when individual-level interventions (i.e., behavioral changes obtained through individualized counselling on lifestyle modifications, psychosocial factors, and weight) are associated with treatment goals and integrated with population-level interventions (namely public health policy and advocacy on specific risk factor interventions, environment, air pollution, and climate change) [2,4,5]. In general, proposed care is currently based on comprehensive therapeutic strategies that are effective on relevant clinical outcomes, including the use of CV drugs that are recommended by current international guidelines [2,4]. Elevated levels of plasma low-density lipoprotein cholesterol (LDL-C) play a crucial role in the development of CAD; the recommended drugs are statins, selective cholesterol absorption inhibitors (e.g., ezetimibe), and the most recent proprotein convertase subtilisin/kexin type 9 inhibitors (PCSK9i), variously combined [2]. Diabetes is another relevant, independent risk factor for CAD, and its appropriate management can reduce the risk of micro- and macro-vascular outcomes and related mortality [6]. Elevated blood pressure is a major risk factor for CAD [1,2]. The combination therapy is based on blood pressure level and total CAD risk; diuretics, angiotensin-converting enzyme inhibitors (ACE-is), angiotensin receptor blockers (ARBs), calcium antagonists, and β-blockers are all recommended [1,2,7,8]. Antiplatelets significantly reduce the occurrence of clinical events for CAD; acetylsalicylic acid and P2Y12 inhibitors can be variously prescribed based on a case-by-case balance between ischemic benefit and the risk of bleeding [1,9], whereas oral anticoagulants are only indicated in specific cases [4,10]. Finally, several anti-inflammatory drugs are under evaluation to prevent the progression of coronary atherosclerosis and related atherothrombotic events (e.g., colchicine and canakinumab) [6]. By implementing current guideline recommendations in clinical practice, the benefit to the patient will likely be maximized. Moreover, since a patient with CAD often presents more than one traditional CV risk factor, a multidisciplinary approach between cardiologists, other specialists, and general practitioners (GPs) is essential [3,4]. Internationally published real-world analyses have clearly shown that clinical practices remain inconsistent with international guidelines [11,12,13].

This study of Italian healthcare administrative databases is aimed at describing patients with CAD and eligible for secondary CV prevention and assessing their healthcare resource consumption and costs from the perspective of the Italian National Health Service (INHS).

## 2. Materials and Methods

### 2.1. Data Source

This analysis was performed by means of healthcare administrative data from the Fondazione Ricerca e Salute (ReS) database, which has previously been used for several observational studies on various clinical questions in different clinical fields since 2018 [14,15,16]. Fondazione ReS is a non-profit foundation with the purpose of creating useful tools for planning and monitoring healthcare policy issues at various levels and for different stakeholders. This study originated from the record linkage of the healthcare administrative databases owned by several Italian Local Health Units (LHUs) and Regional Health Authorities (variously distributed from north to south), and routinely collected in the ReS database under specific agreements. These databases are periodically sent to the Italian Ministry of Health for reimbursement purposes. The INHS is a universal coverage single-payer healthcare system, thus the healthcare data collected by local and regional databases could potentially represent the healthcare of all the beneficiaries of Italy. Therefore, the ReS database can be considered a reliable representation of the Italian population based on a comparison with the Italian Institute of Statistics (ISTAT) by age group (Appendix A). The demographic database contains age, gender, LHUs of residency, and disease exemption codes for INHS cost sharing. The pharmaceutical database consists of all the drugs reimbursed by the INHS and supplied by both local and hospital pharmacies. Active substances could be analyzed based on marketing code (AIC), Anatomical Therapeutic Chemical code (ATC classification), dose, package number, and dispensing date. The hospitalization database was analyzable through in-hospital diagnoses and procedures (according to the current Italian version of the International Classification, 9th version, Disease Clinical Modification (ICD-9-CM), 2007) deriving from hospital discharge forms of both ordinary and daily hospitalizations [17]. The ICD-9-CM, based on the World Health Organization’s Ninth Revision (ICD-9), is the official system of assigning codes to diagnoses and procedures associated with hospital utilization in the United States. The outpatient specialist care database (containing examinations, diagnostics, and invasive/non-invasive procedures performed in an ambulatory setting), supplied by the INHS, was analyzable based on the current national classification system. In addition, all healthcare resources’ databases comprised the costs paid by the INHS in relation to specific healthcare utilization. The record linkage was possible thanks to a unique, anonymized identity number. Indeed, based on the rules on privacy, demographics are completely anonymized at the source. The ReS database was physically placed into Cineca supercomputers. The collaboration with this Italian facility guaranteed compliance with international standard certifications of quality and safety of data management. Finally, since this study was founded on the reuse of anonymous administrative data and conducted for institutional purposes, in agreement with Italian health facilities (Local and Regional Health Authorities), ethical approval was not sought.

### 2.2. Cohort Selection

All patients older than 35 in the ReS database in 2018, and with at least a healthcare resource consumption in the charge of the INHS since 2015, defined the starting population. Among them, subjects who were admitted to hospital at least once in 2018 (accrual) and whose hospital discharge contained a primary/secondary diagnosis of CAD and/or a procedure on coronary arteries, or patients with a CAD-specific cost sharing exemption code, were selected for the analysis (for code descriptions, see Appendix A). Analyses were only carried out on patients suitable for long-term secondary prevention treatments. Therefore, subjects with end-stage diseases, such as those on dialysis therapy or affected by neoplasia (Appendix A), were excluded [2]. The latest date identifying CAD (hospitalization with CAD diagnosis or the assignment of a CAD-specific cost sharing exemption code) in 2018 was considered as the index date.

### 2.3. Analyses

#### 2.3.1. Demographics and Clinical Characteristics

At the index date, patients were characterized in terms of age and gender.

In the analyzable period prior to the index date (specifically, going back to 2015), the most relevant comorbidities were identified (for selection criteria, see Appendix A), these included: heart failure, atrial fibrillation, cerebrovascular diseases, depression, diabetes, dyslipidaemia, arterial hypertension, rheumatoid arthritis, chronic liver diseases, and chronic lung diseases.

#### 2.3.2. Pharmacological Treatments

The analysis cohort was followed up 1 year after the index date in order to assess the consumption of drugs reimbursed by the INHS. These were analyzed after being split into:−Therapeutic strategies specific to secondary prevention: at least a free, filled prescription of ACE-is (plain and their combinations (ATC codes: C09A/C09B), ARBs (plain and their combinations (C09C/C09D), β-blockers (C07), lipid lowering agents (C10), and antiplatelet agents excluding heparin (B01AC).−Other CV drugs: at least a dispensation of pharmacological treatments belonging to ATC codes C and B01, different from those previously mentioned.−Non-CV drugs: at least a free, filled prescription of the drugs identified by ATC codes, different from those quoted in the above bullets.

Pharmacological treatments were assessed by different ATC code levels in terms of the portion of patients treated and mean DDD (a standard measurement of the average daily consumption of the maintenance dosage) per treated subject.

#### 2.3.3. Hospitalizations

Hospitalizations were analyzed by means of the information included in the hospital discharge form. They were analyzed in terms of the primary diagnoses of:−Relevant CV diseases: acute coronary syndrome (ACS), angina pectoris, heart failure, haemorrhagic stroke/intracranial bleeding, ischemic stroke/transient ischemic attack (TIA).−Other CV diseases.−Non-CV diseases.

Selection codes identifying all the aforementioned diagnoses are available in Appendix A.

#### 2.3.4. Healthcare Integrated Costs

Each healthcare administrative database also contained expenditure information. In particular, the healthcare costs analyzable by means of the ReS database were:−Pharmaceutical costs extrapolated by the gross expenditure of local pharmacies’ sales and by the real hospital price (inclusive of value-added tax) of hospital pharmacies’ supplies.−In-hospital expenses derived from DRG (Diagnosis-Related Group) tariffs. Each hospital discharge form was linked to a DRG code which synthetizes the entire healthcare provided at admission and throughout the in-hospital stay.−Outpatient specialist care costs derived from the current National system tariffs.

Findings were provided as mean annual cost per capita of the single items (pharmaceutical, in-hospital, outpatient specialist care) and of their integration (overall cost).

#### 2.3.5. Statistical Analyses

Generally, when administrative data are analyzed, the number of patients/events is so large that even minimal differences will result in a conventional level of statistical significance, often without a corresponding and convincing level of clinical significance. For this reason, we have mostly avoided the use of detailed *p* values and have described nominal differences.

All analyses were performed by means of the Oracle SQL Developer, Italian version 18.1.0.095, California, United States.

## 3. Results

### 3.1. Demographics and Clinical Characteristics

For more than five million inhabitants that were analyzable through the ReS database in 2018, 46,063 patients aged ≥ 35 (1.3%) were found to be affected by CAD and suitable for long-term secondary CV prevention treatments, according to the selection criteria described in the Section 2 (Figure 1).

Males (72.1%) made up most of the CAD cohort, and the mean age (±SD) was 70 ± 12 (Table 1). The prevalence of patients with CAD increased with age, with a peak at the 75–84 age group (Table 1), for both males and females.

### 3.2. Pharmacological Treatments

During the one-year follow-up, 96.4% of the cohort was treated with at least one drug for CV secondary prevention, with a mean consumption per patient of 1262 DDD (Table 2).

### 3.3. Hospitalizations 

Exactly 30.6% of the cohort was hospitalized at least once during the one-year follow-up (Figure 2).

In addition, 11.4% of patients were hospitalized due to relevant CV events. They were most frequently admitted due to ACS (5.2%) and heart failure (4.0%). Heart failure and haemorrhagic stroke/intracranial bleeding caused the longest in-hospital period of stay (approximately 11 days), as shown in Appendix A. Furthermore, 9.1% of the cohort was hospitalized due to other CV events. A great percentage of patients (17.5%) was admitted for non-CV events, mostly due to diseases of the lung (acute pulmonary infections, lung diseases due to external agents, asthma, and chronic obstructive pulmonary diseases), and the replacement of organ or tissue. Further details are shown in Appendix A.

### 3.4. Healthcare Integrated Costs

On average, during the year following the index date, each patient with CAD and eligible for secondary CV preventive treatments cost the INHS EUR 6078 (Figure 3).

The overall expenditure originated from the integration of all the health care charged to the INHS. Hospitalizations accounted for 69.9% of the overall expenditure. A substantial contribution was provided by non-CV causes (25.2% of the hospitalization expenditure). Pharmaceuticals accounted for 24.7% of the overall cost. Concomitant non-CV drugs, which also accounted for 11.7% of the overall expenditure per capita, determined approximately half of pharmaceutical costs.

## 4. Discussion

### 4.1. Demographics and Clinical Characteristics

This study has assessed a population aged ≥35, affected by CAD, and eligible for secondary prevention, through Italian healthcare administrative data. The burden of CAD and the preventive strategies for further CAD risk reduction are, overall, not negligible, and increase with age [18,19]. Age and gender distributions from this analysis are in line with expectations and other studies [20,21,22].

Patients affected by CAD have several comorbidities (approximately 50% of them by three or more), which are also associated with the risk of further CV diseases. In this study, patients with CAD and eligible for secondary prevention were mostly affected by arterial hypertension, dyslipidaemia, and diabetes; this is in line with other real-world studies [3,11,20]. The aim of secondary preventive therapy of CAD is to slow down the progression of the atherosclerosis and prevent its severe clinical manifestations. This can be reached by means of the integration between the non-pharmacological management of risk factors (i.e., lifestyle counselling, especially smoking cessation, dietary changes, and weight reduction) and specific pharmacological treatments for both traditional, modifiable risk factors and comorbidities [1,2,9,10].

### 4.2. Pharmacological Treatments

Despite the prescription rate of preventive drugs having been on the increase for many years [22], and the extensive documentation and recommendations of secondary preventive strategies [11], prescription rates and the persistence of preventive treatments have frequently shown to be suboptimal and not fully compliant with current guideline recommendations [11,12,13,20]. In particular, it has been demonstrated that patients hospitalized for MI (the most frequent manifestation of CAD [11]) reduced their drug coverage after one year following their discharge [22]. With the purpose of describing the complete medication pattern, we split pharmaceutical supplies into CV drugs for secondary prevention, other CV drugs, and non-CV drugs. From the analysis of the specific CV drugs supplied during one year after the index date, at least one of them was dispensed to 96.4% of patients; antiplatelets were the most supplied, followed by lipid lowering agents, β-blockers, ACE-is, and ARBs. This was found to be consistent with the findings provided by Shimada and colleagues [21]. However, as real-world studies on this condition are still few, and healthcare policies among countries are inevitably heterogeneous, a reliable comparison is not possible [22]. Regarding our study, we only selected and analyzed subjects with CAD and eligible for secondary preventive therapies, and, consequently, very few patients (3.6%) did not receive any specific CV drug. The causes for non-prescription could include the transfer to private or residential facilities, a Local or Regional Health Unit not being covered by the ReS database, and the private purchase of therapies or the refusal of them. Through administrative databases, real consumption is not fully evaluable. Some under-prescriptions can be justified by the prevalently older age that characterized this target population. Older age is an independent predictor of worse adherence due to the high number of comorbidities and polypharmacy, reduced mobility (to reach physicians and pharmacies) and cognitive ability, and the financial issues to obtain drugs (such as in countries where healthcare is mainly based on health assurance capitals). At the same time, under-prescriptions could also be due to the underestimation of CV risk in women, and, more generally, to contra-indications or adverse events which might outweigh the expected benefits, or the lack of knowledge about a patient’s diagnosis [21,22].

The prescription pattern of non-CV drugs reflects the clinical conditions of an elderly population, which is frequently treated with drugs for gastric disorders and anti-inflammatory or antibacterial agents, even in the absence of evidence-based recommendations.

Different solutions have been proposed to favor the compliance to drug recommendations and improve the control of the CAD progression. Shimada and colleagues, by demonstrating that drugs are rarely prescribed by GPs after hospital discharge, have suggested that an improvement in prescriptions at discharge may be one of the most important interventions [21]. Supporting this concept, Jortveit and colleagues found that patients with MI were less likely to be prescribed with secondary preventive therapy at their hospital discharge compared to non-prior CAD subjects [11]. On the contrary, Ulrich and colleagues, by considering the significant reduction in secondary prevention treatments at one year after discharge for MI, recognized that the GP ambulatory is the first setting in which the secondary prevention can be improved [22]. However, in general, it is crucial to increase the awareness of the level of risks in patients, caregivers and practitioners, and to train and update healthcare professionals about the guidelines’ recommendations [12] due to the dynamism of CV risks which are manageable with ever-novel therapies [3].

### 4.3. Hospitalizations

The risk of recurrent admissions at one year after a CV event, especially for MI and stroke, has been found to be high, specifically for patients who did not follow guideline-recommended therapies [3,11]. The persistent risk of CV diseases in patients with chronic CAD has been shown by different studies [3], which, for example, have found that a one-year incidence of major adverse CV events (i.e., stroke, MI, and CV death) was 4.5% to 20.7% for the overall CAD population [3,20]. Our study shows that the rate of patients hospitalized during the one-year follow-up due to relevant CV events (i.e., ACS-included MI, heart failure, angina pectoris, ischemic stroke/TIA, and haemorrhagic stroke/intracranial bleeding) was 11.4%, which is a not negligible rate. A high proportion of subjects was hospitalized due to a new CAD diagnosis (5.2% of subjects with ACS), with 1.4% due to ischemic stroke/TIA. Interestingly, these findings are consistent with the results of a French claims database analysis, which provided that, during the one year following a first hospitalization for MI, 3.1% of individuals experienced a new MI and 1.9% experienced stroke/TIA [20]. All these data reflect an unsatisfactory risk factor control and a high rate of new atherothrombotic events in patients with prior CAD.

Exactly 9.1% of our cohort was hospitalized due to other CV events, however a relevant rate (17.5%) of patients was admitted due to non-CV causes, once again reflecting the elevated mean age of the target population and the burden of comorbidities. Moreover, 30.6% of our cohort was admitted to hospital at least once for any cause during the one-year follow-up. This rate of readmissions is close to that found by a German claims database analysis, which reported that from the third to the fifth quarter after MI, 40% of all patients were re-hospitalized due to any condition [22].

### 4.4. Healthcare Integrated Costs

The overall integrated cost charged to the INHS of a patient with CAD and eligible for secondary preventive treatments was elevated. The cost distributions of the overall expenditure clearly reflect the demographic and clinical conditions of this cohort, which was mostly elderly and affected by multiple comorbidities. Interestingly, the outpatient specialist services weighed a little. Causes could be the high utilization of the private outpatient specialist resources or the scarce frequency of controls and visits, especially for the cardiological ones. Interestingly, Ulrich and colleagues found that, one year after MI, only 22.8% of patients was visited by a cardiologist, only 34.1% received a laboratory investigation, and the reduction in visits to the cardiologist was particularly high in patients aged from 65 to 86 [22].

Cost-effectiveness has been found to be strongly dependent on parameters such as the age of the target population, the overall population risk of CV diseases, and the interventions’ cost [2]. Delate and colleagues retrospectively evaluated the financial impact of a comprehensive cardiac care service program from the perspective of their health plan [23]. They found that when patients were assisted by a collaborative cardiac care service (a multidisciplinary program co-managed by clinical pharmacy, specialist, and nurses), inpatient and outpatient hospitalizations, and medical office visits, were associated with lower healthcare resource utilization and costs. They therefore encouraged substantial investments in early treatment initiation, aggressive drug dosage titration to achieve treatment goals, and long-term adherence to evidence-based secondary prevention treatment recommendations in order to achieve cost-offsets both in cardiac-related and in overall healthcare.

### 4.5. Strengths and Limitations

The perspective was only that of the INHS, both for the in-hospital and outpatient settings, therefore, the healthcare services not reimbursed by the INHS or that were privately provided (e.g., for private purchase or specialist care) were not collected and analyzed in the administrative databases, leading to an inevitable underestimation. In particular, the Italian Medicine Agency reported that, in 2018, 23% of overall pharmaceutical expenditure was addressed to private purchase [24]. Nevertheless, claims databases have been considered reliable in identifying populations affected by chronic diseases, with limitations mostly due to differences in healthcare performance among medical conditions, and in obtaining evidence on policy impact in a low-cost, timely, and reproducible manner [25,26,27]. Moreover, the ReS database can be considered reliably representative of the Italian population when comparing its age distributions with those of the Italian Institute of Statistics (Appendix A). Laboratory values were not collected, thus the treatment goals were not available and analyzable. It was not possible to infer causality between the use of CV drugs or the prevalence of comorbidities corresponding to CV risk factors and one-year hospitalization due to CV causes. The appropriateness of prescription was not evaluable due to the lack of clinical information (e.g., diagnosis or severity of the disease, which justify the drug prescription). This study only described the healthcare costs that were charged to the INHS, focusing on free, filled drug prescriptions, hospitalization, and costs, without being able to also include personal lifestyles and social burden, since they are not recorded in administrative databases. However, even if this study described only a part of the secondary CV prevention care pathway in Italy, it provided real-world evidence of the healthcare and economic burdens and criticalities for stimulating the improvement in a still suboptimal management of the secondary prevention. Finally, thanks to the coverage of a very large and heterogeneous population, thus more representative of the real population affected by this clinical condition than clinical trials or specialty registries, and to a long-term follow-up of filled drug prescriptions, outpatient specialist visits, procedures, and hospitalizations, the analysis of the ReS database provided updated and complementary information, which can be generalized to the entire population of beneficiaries.

## 5. Conclusions

This is the most recent study on Italian healthcare administrative data, specifically designed for describing the epidemiology of patients with CAD, their public healthcare (i.e., exclusively reimbursed healthcare), including the specific pharmacological secondary prevention and costs covered by the INHS. Moreover, the present analysis can be considered complementary to clinical studies or registry studies since it provides updated information generalizable to the total population of subjects with this clinical condition. This analysis of the ReS database shows that patients with CAD determine a relevant burden for the INHS. The age of these patients is generally elevated, with very frequent comorbidities. This fact is particularly clear when non-CV reasons for hospital admission and the costs for non-CV drugs are considered. This study also shows that secondary prevention therapies remain insufficiently used in a real-world setting, highlighting a potential under-prescription of the guideline recommended drugs for effective CV risk factor control. Areas of improvement are various, and found at patient, physician, and population level. The healthcare of these patients, as well as the burden on the health system (healthcare resource utilization and costs), would benefit from a collective and multidisciplinary effort on the part of all stakeholders involved in the care pathway.

## Figures and Tables

**Figure 1 jcm-10-04708-f001:**
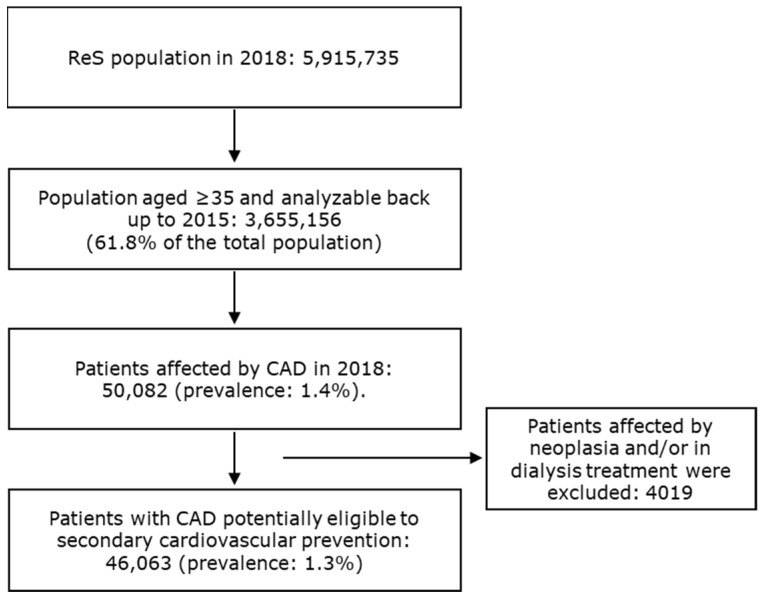
The flowchart describes the selection of patients affected by CAD and eligible for secondary cardiovascular prevention.

**Figure 2 jcm-10-04708-f002:**
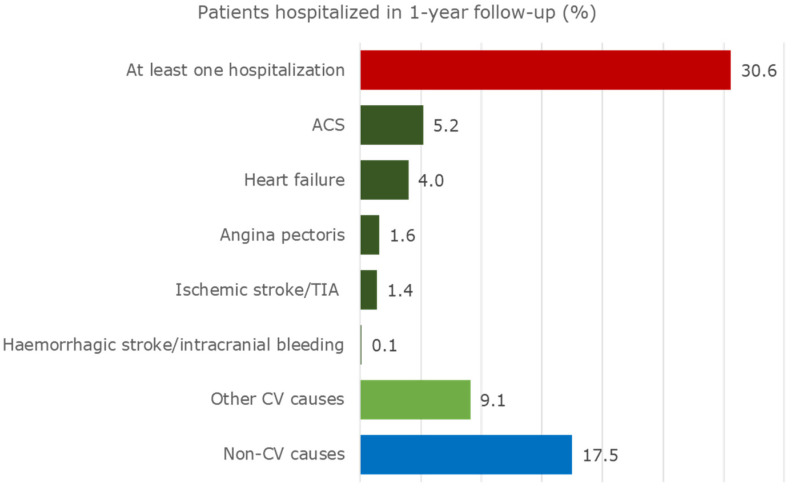
Hospitalizations of patients with CAD and eligible for CV secondary prevention drugs during the one-year follow-up.

**Figure 3 jcm-10-04708-f003:**
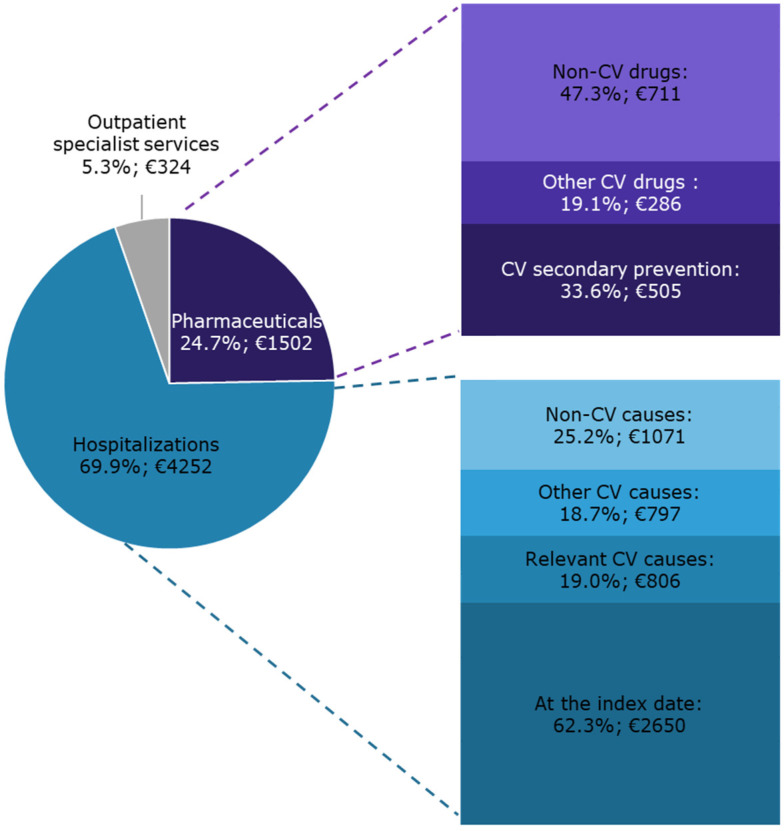
The INHS healthcare costs of patients with CAD and eligible for CV secondary preventive treatments during one-year follow-up are shown as the distribution of the specific cost items and as mean expenditure per capita (€).

**Table 1 jcm-10-04708-t001:** Characterization of the cohort of patients affected by CAD and eligible for long-term secondary prevention treatments in 2018 among ReS population aged ≥ 35, in terms of demographics at the index date and of comorbidities in the available previous period.

Demographics and Clinical Characteristics	Patients with CAD and Eligible for Secondary Prevention (*n* = 46,063)
Males (n; %)	33,234; 72.1
Median age (Q1; Q3)	70 (60; 83)
Mean age (±SD)	70 ± 12
Distribution by age group (prevalence × 1000 subjects aged ≥ 35)	
35–44	0.9
45–54	4.7
55–64	13.9
65–74	23.8
75–84	27.3
≥85	21.3
Overall cohort	12.6
Comorbidities (n; %)	
None	1858; 4.0
1 comorbidity	5361; 11.6
2 comorbidities	15,646; 34.0
3 comorbidities	12,862; 27.9
4 or more comorbidities	10,336; 22.4
Arterial hypertension	41,415; 89.9
Dyslipidaemia	33,332; 72.4
Diabetes	15,276; 33.2
Chronic lung diseases	9645; 20.9
Heart failure	7180; 15.6
Atrial fibrillation	5641; 12.2
Depression	4803; 10.4
Cerebrovascular diseases	2992; 6.5
Chronic liver diseases	1875; 4.1

The analysis of comorbidities showed that 96.0% of the cohort was affected by at least one concomitant disease, while approximately 50.0% was affected by three or more. Specifically, our cohort of patients mostly suffered from arterial hypertension (89.9%), dyslipidaemia (72.4%), and diabetes (33.2%).

**Table 2 jcm-10-04708-t002:** Pharmacological treatments supplied to patients with CAD and eligible for cardiovascular (CV) secondary prevention drugs during the one-year follow-up.

Supplied Drugs	Patients Treated (n; %)	Mean DDD Per Treated Patient
Drugs for CV secondary prevention (in descending order)		
Antiplatelet agents (excluding heparin)	38,201; 82.9	400.8
Lipid lowering agents	38,042; 82.6	532.1
β-blockers	33,375; 72.5	136.8
ACE inhibitors (ACE-is)	20,964; 45.5	469.4
Angiotensin II receptor blockers (ARBs)	15,564; 33.8	388.4
At least one drug for CV secondary prevention	44,391; 96.4	1261.6
Concomitant CV drugs (first 10 supplied, in descending order)		
Loop diuretics	18,239; 39.6	300.4
Antithrombotic agents (excluding antiplatelets)	11,439; 24.8	173.7
Selective calcium channel blockers with mainly vascular effects	8736; 19.0	320.4
Potassium-sparing agents	7351; 16.0	126.5
Other cardiac preparations	7271; 15.8	202.5
Nitrates	5426; 11.8	338.2
Anti-arrhythmics, class I and III	4340; 9.4	196.5
Anti-adrenergic agents, peripherally acting	2126; 4.6	198.7
Diuretics and potassium-sparing agents in combination	1920; 4.2	95.8
Selective calcium channel blockers with direct cardiac effects	1547; 3.4	201.5
At least one concomitant CV drug	31,948; 69.4	518.3
Concomitant non-CV drugs (first 10 supplied, in descending order)		
Drugs for peptic ulcer and gastro-oesophageal reflux disease	36,520; 79.3	230.0
Anti-inflammatory and antirheumatic products, non-steroids	17,305; 37.6	43.9
Beta-lactam antibacterials, penicillins	15,106; 32.8	14.6
Quinolone antibacterials	12,408; 26.9	13.1
Blood glucose lowering drugs, excluding insulins	12,293; 26.7	359.8
Other beta-lactam antibacterials	10,628; 23.1	9.3
Vitamin A and D, including combinations of the two	10,015; 21.7	454.0
Antigout preparations	9857; 21.4	155.4
Corticosteroids for systemic use, plain	9285; 20.2	60.1
Drugs used in benign prostatic hypertrophy	7646; 16.6	377.1
At least one concomitant non-CV drug	44,120; 95.8	861.2

At least one concomitant CV drug was dispensed to approximately 70% of the cohort (most frequently, antihypertensive and antithrombotic agents), which recorded the lowest mean consumption (518 DDD). More than 90% of patients were supplied with at least one non-CV drug, mostly proton pomp inhibitors (PPIs), anti-inflammatory treatments, and antibacterials for systemic use, and, on average, each patient consumed 861.2 DDD.

## Data Availability

No new data were created or analyzed in this study. Data sharing is not applicable to this article.

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
