# Peer review of "Coronary Artery Disease in Patients Older than 35 and Eligible for Cardiovascular Secondary Prevention: An Italian Retrospective Observational Analysis of Healthcare Administrative Databases"

_jcm, 2021, doi:10.3390/jcm10204708_

Round 1

Reviewer 1 Report

The authors conducted an observational, retrospective study of a CAD patient sample representative of the Italian population. The authors describe a rigorous methodology, however the overall results are repetitive of other real-world studies (for example the multicentre, multinational EUROASPIRE study). The overall paper comes with the limitations of its retrospective design and the flaws of choosing ICD-9 codes for disease definition and morbidity identification.

The current study adds little to the existing literature on CAD and this is evident by the double use of the verb “confirms” in the Conclusions section.

Reviewer 2 Report

The work submitted to me for review is a very interesting study on Coronary Artery Disease in Patients Older Than 35 and Eligible to Cardiovascular Secondary Prevention. The conducted retrospective analysis covers a large group of patients and contains a lot of characterizing information, however, in my opinion, the potential associated with the acquired data was not fully exploited. Additionally, there is no reference to the current guidelines of the European Society of Cardiology. These are the two biggest drawbacks of the manuscript in its current form and need to be addressed.

In addition, I have the following several comments:

  1. INTRODUCTION

  • The authors write about prevention based on the 2016 ESC guidelines; new guidelines on cardiovascular prevention are available since the last ESC congress.

  • The authors when listing hypotensive treatment options are not thorough enough and do not refer to current literature, this also applies to information regarding hypolipemic and antiplatelet treatment e.g. currently recommended regimens for combining antiplatelet and antithrombotic treatment in patients after ASC (TIMI 54 study, Compass study).

  • In the introduction, the authors describe triggers and factors influencing coronary artery disease (CAD) progression. In my opinion, environmental factors such as air pollution should not be overlooked as they are also included in the latest ESC guidelines. There are many papers from Europe on this issue, also from Italy. Here are some examples of recent publications directly related to acute coronary syndromes and air pollution:

  • DOI: 1016/j.ijheh.2020.113578
  • DOI: 10.1016/j.envres.2021.111154

  1. MATERIAL AND METHODS

  • This section lacks information regarding statistical analysis. In my opinion, the authors should be encouraged to compare between groups and perhaps consider building regression models to assess factors influencing lack of compliance with guidelines and/or incidence of hospitalizations and deaths.

  • Heparin is not an antiplatelet drug and, in my opinion, the methodology should list antiplatelet and anticoagulant drugs separately.

  • There is a lack of inclusion of chronic kidney disease as one of the major factors affecting CCS.
  1. RESULTS AND DISCUSSION

  • Patient characteristics lack information regarding their past history with specific reference to previous ACS, which can be obtained from Supplementary Materials.

  • β-Blockers and/or calcium channel blockers (CCBs) are not drugs that affect prognosis but only symptom control in patients with CCS.

  • The literature analysis lacks up-to-date literature on both guidelines and references to other analyses e.g. KOS-ZAWAŁ from Poland.

  • The literature is extremely poor and mostly outdated.

Reviewer 3 Report

Coronary artery disease in patients older than 35 and eligible to cardiovascular secondary prevention: and Italian retrospective observational analysis of healthcare administrative databases

Overall, well-written manuscript with extensive information and analyses.

However, description on current data analyses from healthcare database, with little proposal, suggestion or new insight from the manuscript.

Major comments:

Healthcare burden regarding coronary artery disease patients with multiple comorbidities and hospitalization requirements is widely-known issue in preventive patient care. Need for multidisciplinary approach to improve patients’ outcome and reducing costs is also a well-documented theme. I regretfully cannot find new insights or further suggestion from this extensive descriptive analysis of the current Italian healthcare database.

However, it may have meanings of current analytical data and clinicians may be provided with extensive, but not innovational, information.

If possible, the authors may have benefit from making their efforts more insightful by adding new suggestions or proposal to current practice or preventive medicine.

Round 2

Reviewer 1 Report

No further comments.

Reviewer 2 Report

The authors responded to my suggestions and made corrections. I have nothing more to add.

Reviewer 3 Report

The authors have responded to the reviewer's concern.